# Does China’s National Demonstration Eco-Industrial Park Reduce Carbon Dioxide and Sulfur Dioxide—A Study Based on the Upgrading and Transformation Process

**DOI:** 10.3390/ijerph191912957

**Published:** 2022-10-10

**Authors:** Kairui Cao, Laiqun Jin, Yuanyuan Zhu, Zilong Nie, Hao Li

**Affiliations:** 1Business School, Ningbo University, Ningbo 315211, China; 2Marine Economics Research Center, Donghai Academy, Ningbo University, Ningbo 315211, China; 3Faculty of Foreign Languages, Ningbo University, Ningbo 315211, China

**Keywords:** eco-industrial parks, environmental pollution, upgrading and transformation of development zones

## Abstract

With the increasingly serious environmental problems, coordinating the relationship between the environment and economic development has become a crucial task for developing countries, especially China. This paper studies the role of eco-industrial parks (EIPs) in the emissions of carbon dioxide and sulfur dioxide in China with the difference-in-difference (DID) approach by focusing on the entire process of EIPs’ establishment—upgrading provincial development zones (DZs) to national DZs and then transforming national DZs into EIPs. Besides, we examined the heterogeneous effect of the different transformations from national economic and technological development zones (ETZs) or national high-tech zones (HTZs) to EIPs. In addition, we studied the spatial spillover effects of EIPs and their paths with the spatial difference-in-difference (SDID) method. The results show that neither provincial DZs nor national DZs can significantly reduce sulfur dioxide and carbon dioxide emissions. Only when national DZs are transformed into EIPs can they be reduced significantly. However, the different transformations from the HTZs and ETZs to EIPs have different effects on emissions. Moreover, EIPs have technology spillovers and demonstration effects on surrounding areas. Therefore, EIPs can reduce emissions in the surrounding areas. The results indicate that, in order to achieve high-quality development and coordinate the relationship between environment and economic development, we should take positive steps to promote the transformation of DZs into EIPs.

## 1. Introduction

Since the reform and opening up of China in 1978, the policy of development zones (hereafter “DZs”) has significantly contributed to China’s economic growth through economic agglomeration [1,2,3], with 387 national-level DZs and 2299 provincial-level DZs established by 2021. However, the establishment of DZs mainly brought about extensive economic growth, consumed large amounts of resources and sacrificed the environment in exchange for economic growth with overcapacity, excessive environmental pollution, and high carbon dioxide emissions [4,5], which are detrimental to the sustainable development of the country. Air pollution and greenhouse gas emissions are two essential aspects of environmental problems. Numerous studies have pointed out that air pollution poses a significant threat to human health, including reducing the function of the human immune system [6], affecting the cardiovascular system [7,8], and even causing diseases such as cancer [9]. Meanwhile, greenhouse gas emissions cause many climate changes that are harmful to human activities, such as global warming, droughts, floods, and storms [10]. Some respiratory diseases which do harm to human health are also caused by greenhouse gas emissions [11]. Therefore, how to change the development mode to harmonize the relationship between environment and economy has become an important issue.

In order to achieve sustainable development and change the extensive economic growth mode, the State Environmental Protection Administration of China (SEPA) launched the pilot project of National Demonstration Eco-industrial Parks (hereafter “EIPs”) in 2001. The approval of the construction of Guangxi Guigang National Demonstration Eco-industrial (Sugar) Park in 2001 marked the official start of the construction of China’s EIPs. By 2021, a total of 55 EIPs had been established, and 45 EIPs have been approved for construction as well [12,13]. The eco-industrial park is a new type of industrial park designed and established according to the requirements of clean production, the concept of circular economy, and the principle of industrial ecology [14]. Its main purpose is to connect different factories or enterprises by means of logistics or energy flow transfer, forming a symbiotic combination of industries that share resources and exchange by-products [15], so that waste or by-products of one factory can become raw materials or energy for another factory [16,17]. It can simulate the natural system in the park and establish the “producer-consumer-decomposer” cycle in the industrial system. With the goal of creating a high-tech, high-efficiency and environmentally friendly park, it can also realize the closed-loop circulation of materials, the multi-level utilization of energy, and the minimization of waste generation [17,18,19].

Then, how does the establishment of EIPs affect the environment? Previous studies have focused on the agglomeration economy and circular economy of EIPs, which attract capital and human resources through various financial subsidies and preferential taxation to promote technological innovation and industrial upgrading. The agglomerations of industries that meet the entry threshold of EIPs are mostly high-tech and low-pollution industries. Besides, the park has strict environmental assessment and supervision, making the park pay attention to environmental issues while developing the economy [20]. In this way, the green innovation capability of enterprises can be significantly improved [20,21,22]. Not only that, these agglomerations may also form a network of energy symbiosis networks which promotes the cooperation and efficient sharing of resources among enterprises within the industrial agglomeration. Through energy exchange and energy synergy effects, it can greatly improve resource utilization efficiency at the same time [23,24]. In addition to its agglomeration economy, the park has built a new circular economy system which improves resource utilization efficiency by promoting the recycling of inter-industry waste, reducing the use of resources and the emission of pollutants in the park [25]. For example, the recycling of wastewater minimizes the utilization of freshwater resources in the park and reduces the discharge of wastewater from the park [26]. The incineration of solid waste can be saved for power generation. Not only China, but also other countries around the world have made attempts to reconcile economic growth and the environment. In order to protect the environment in the process of development, some scholars have evaluated the effects of EIPs in various countries [27,28,29,30]. Kim (2018) [31] finds that the establishment of EIPs in South Korea not only reduces environmental pollution, but also improves the overall economic performance of upstream and downstream industries throughout the supply chain. Abu-Qdai (2022) [32] used a life cycle assessment (LCA) to analyze the environmental performance of one of the EIPs in Russia, showing that these enterprises are more sustainable than the business-as-usual scenario. The EIPs also significantly reduce the environmental impact of the production process.

Most previous studies have directly and independently analyzed the environmental impacts of establishing EIPs and their impact pathways. Nevertheless, China’s EIP policy has a prominent feature—upgrading and transformation. It is not a stand-alone policy, previous studies have ignored this feature, which may cause biases in the results. For example, Jiang (2022) pointed out DZs increase economic growth while boosting pollution and that HTZs can reduce pollution. EIPs are upgraded and transformed by DZs [33], ignoring this process and analyzing the effect of EIPs directly. The results obtained are the joint effect of DZs and EIPS rather than the effect of EIPs. Also, it can’t be ascertained whether the final reduction of pollution is attributed to EIPs or HTZs. Before the establishment of a provincial DZ, an application must first be made to the provincial administrative planning or science and technology department to obtain approval. After the provincial DZ has reached a certain scale, it can be upgraded to a national-level DZ by applying to the relevant central ministries and commissions, usually to the Ministry of Science and Technology for a national high-tech zone, (hereafter “HTZ”) and to the Ministry of Commerce for a national economic and technological development zone (hereafter “ETZ”), and finally to the State Environmental Protection Administration for an EIP to achieve green development [32]. When the impact of the upgrading process on the environment is ignored, the estimation results will be biased by the combined effect of the upgrading of provincial DZs to national DZs and the transformation of national DZs to EIPs. Firstly, if the transition process of DZs to EIPs is ignored, and the impact of DZs on the environment is generally analyzed [33], then the effect of DZs on the environment will be biased because the effect of the EIPs is included. Similarly, if only cities without EIPs are used as the control group, and the effect of EIPs is only simply analyzed, then the effect of EIPs on the environment will be biased because the effect of the upgrading to national DZs is included. Therefore, this paper makes up for the defects of previous studies by considering the entire upgrading and transformation process. The impact of EIPs establishment on sulfur dioxide and carbon dioxide emissions is more precisely indicated through this paper.

Meanwhile, China’s policies are often carried out step by step, like in the building of EIPs, where a percentage of cities are first selected to establish EIPs, while a corresponding percentage of other cities will not have the right. Such a process will lead to inter-regional flows of companies attracted to areas where EIPs are established, and thus EIPs will have an impact not only on local sulfur dioxide and carbon dioxide emissions, but also on emissions from surrounding areas [34,35]. Preferences and subsidies for EIPs will attract enterprises or resources from areas which without EIPs to areas with EIPs, and such spatial flows will also affect pollutant emissions distribution among cities [12]. Because EIPs are concerned about the environment and prefer green high-tech industries, most of the enterprises attracted to EIPs may have high technology and low pollutant emissions [30], which inevitably reduce the local pollutant emission intensity, while the neighboring areas have higher pollution emission intensity due to the relocation of such enterprises out of the area. At the same time, there are synergistic effects between the EIPs and the neighboring areas [16] and there are technological spillover and demonstration effects from the EIPs to the neighboring areas, which will also affect the pollutant emissions in the neighboring areas. Based on this, we use the spatial DID method to analyze the impact of EIPs on sulfur dioxide and carbon dioxide emissions in the surrounding areas.

Finally, this paper proves that the establishment of EIPs can improve the local technology level as a way of reducing local pollutant emissions and reducing the emissions of pollutants in the surrounding area. In summary, the marginal contributions of this paper are that, unlike the previous studies which mainly focused on the impact of EIPs on the environment alone, we focus on the whole process of the upgrading and transformation of EIPs. We will also correct the previous assessment of the environmental impact of DZs, because they all mistakenly included the role of EIPs in the role of DZ. Meanwhile, the spatial DID method is applied to calculate the reconfiguration effect of EIPs on air pollution to judge whether the establishment of EIPs will impact the emissions of the surrounding areas while affecting the local. This paper proceeds as follows: the policy background of EIPs is introduced in Section 2. The empirical methodology of DID used in this paper is discussed in Section 3. In the next section, the empirical results of this paper are reported, and the results are explained. In Section 5, the results of the heterogeneity analysis of ETZs and HTZs transformed into EIPs are presented. The results of the parallel trend test are presented in the next section. In Section 7, the spatial spillover of EIPs is reported, as well as the mechanism of EIPs affecting the environment through mediating effects in Section 8. The statement of conclusion is in the final Chapter.

## 2. Background

The EIP in China is a demonstration zone that takes pollution prevention as the starting point and is characterized by material circulation flow, with the ultimate aim of social, economic, and environmentally sustainable development. The guiding ideology is to combine the development of the circular economy while giving full play to regional comparative advantages and improving market competitiveness from the height of sustainable development to promote a fundamental change in the production and consumption patterns of the social economy. The basic principle is that the planning and establishment of demonstration zones, follows the behavioral principles of “reduce, reuse, and recycle” [36]. (1) The reduction principle is that the amount of material entering the production and consumption process must be reduced, i.e., fewer raw materials and energy inputs to meet the established production or consumption needs should be used so that resources can be saved and pollution can be reduced at the source of economic activities. In production, it is often required to miniaturize the volume of products, lighten the weight of products, and pursue simple packaging rather than over packaging; in daily life, in order to reduce waste emissions, people’s excessive demand for goods should be curbed. (2) The reuse principle is that products and packaging are supposed to be reused in their initial form. In production, manufacturers are often required to use standard sizes for design to replace parts instead of replacing the entire product while encouraging the development of remanufacturing industries; in life, people are encouraged to buy items, beverage bottles, and packaging that can be reused. (3) The principle of recycling requires that the products should be able to be turned into usable resources instead of useless garbage after completing their valuable functions. There are usually two ways of recycling materials: one is the formation of the same product as the original after resource recycling; and the other is the formation of a different new product after resource recycling. The principle of recycling requires consumers and producers to purchase products with a large proportion of recycled materials to close the whole circular economy process. Hong (2020) points out that EIPs can reduce resource consumption, pollution, GHG emissions, and waste generation, potentially by pre-treating waste for reuse or recycling. It should not be overlooked that companies use recyclable materials to save costs and improve economic benefits. However, environmental protection should also be taken into consideration in the use of recyclable materials [37].

China’s first EIP was established in 2008, and a total of 55 EIPs were established by 2021. The number of national eco-industrial demonstration parks built each year is very scarce, and most of them are concentrated in eastern cities. A total of 17 EIPs were built between 2008 and 2012, completing the initial attempt to transform DZs to EIPs. The country followed the concept of green development henceforth and paid more and more attention to the environment in the process of development. As of 2021, the country has established 55 EIPs. In addition, the policy of DZs has its own upgrade process, from no DZs to the establishment of provincial DZs and then upgraded to national DZs. The evolutionary distribution is shown in the Figure 1.

In 2008, 244 cities in China established Provincials DZs, while 66 cities established National DZs and two cities established EIPs. In order to develop its economy, China established a large number of DZs. At the same time, China began to pay attention to environmental protection and started the transition from National DZs to EIPs. In 2012, 170 cities in China established Provincials DZs and 11 cities completed the transition from National DZs to EIPs, most of which were in the eastern coastal cities. The number of cities with National DZs also increased to 166, while by 2016, the number of cities with EIPs increased to 28, which also concentrated in coastal areas, with 134 Provincial and 205 National DZs. By 2020, 32 cities have established EIPs and are gradually establishing EIPs in the central city. 214 cities have established National DZs, and 125 cities have established Provincial DZ. It can be seen that Provincial DZs are upgrading to National DZs, while National DZs are transforming into EIPs.

In the approval process, the construction of EIPs must be created by the construction unit to the provincial environmental protection department to create an application; after its review and reporting to the State Environmental Protection Administration, the EIPs creation application agreed by the State Environmental Protection Administration, the EIPs construction unit can organize the preparation of the demonstration zone planning, after a period of construction pilot, the construction unit can be submitted to the provincial environmental protection department for the naming of the EIPs, and its submission to the State Environmental Protection Administration. State Environmental Protection Administration reviewed the submitted materials and reported to the General Administration Council for approval. Qualified applicants are approved as demonstration areas and granted a unified sign specification. The development and problems of EIPs construction should be reported quarterly to SEPA demonstration area at the end of the year. At the same time, the latest information on the development of the EIPs should be included in the form of the “circular economy and eco-industry newsletter,” in the construction and approval process of the State Environmental Protection Administration. The National Cleaner Production Center hosts the Circular Economy and Eco-Industry Newsletter. At the same time, SEPA organizes expert groups to conduct regular assessments and inspections of the EIPs and gives commendations and awards to those with remarkable achievements. Referring to the “National Eco-Industrial Demonstration Zone Management Measures,” the assessment is based on the following indicators: (1) Economic development indicators: indicators of economic development level; indicators of economic development potential. (2) Eco-industrial characteristics indicators: the presence of a mature eco-industrial chain; indicators of reuse; indicators of flexibility characteristics; indicators of infrastructure construction. (3) Eco-environmental protection indicators: indicators of environmental protection; indicators of environmental performance Ecological construction indicators; ecological environment improvement potential. (4) Green management indicators include policy and regulation system indicators; management and awareness indicators. Eco-environmental protection indicators include emission coefficients of major pollutants, annual average reduction rate of carbon dioxide emissions, solid waste emissions per unit of industrial value added, total emissions of key pollutants, implementation rate of clean production audits of key enterprises, etc. It can be seen that: when it comes to economic development, the approval of EIPs focuses nowadays more and more on environmental indicators, so the conditions for DZs to become EIPs are quite strict.

In the new stage of development, in order to achieve sustainable development, the national cadre assessment indicators not only focus on economic growth, but also pay more attention to environmental protection. Compared with ordinary DZs, there are more advantages after being transformed into EIPs. First, in terms of national policy support, while enjoying the original preferential policies, the Ministry of Commerce will promote relevant financial institutions to provide financing support for the ETZs that create national eco-industrial demonstration parks, focusing on supporting and promoting EIPs to develop energy conservation international cooperation in environmental protection. Secondly, enterprises in the EIPs will be guided to make full use of the state’s preferential tax policies, and increase support for the EIPs in terms of capital, investment attraction, foreign economic and technological cooperation, and services, etc. Social capital and foreign investment will be encouraged to invest more in EIPs [38]. Thirdly, local governments will also set up special funds to support the development of EIPs, provide targeted financial subsidies for the parks created by EIPs, such as subsidies, subsidized loans, or tax relief [39,40]. They will also focus on increasing policy research and critical projects for EIPs. Moreover, the number of ecological zones is scarce, and the application process is rigorous. It is a scarce label which can attract more capital inflows and enterprises to enter the park to promote economic growth and technological upgrading in the park.

Figure 2 shows the process of upgrading and transforming DZs into EIPs. It should be noted that, there is a distinction between HTZs and ETZs in the process of upgrading provincial DZs to national DZs. The main goal of HTZs is to develop knowledge-intensive and technology-intensive high-tech industries, which mainly include three significant fields: information technology, biotechnology, and new material technology. In contrast, the ETZs aim to increase the total economic volume, with foreign investment pulling and industries mainly in manufacturing and processing industries. Moreover, from declaration to approval, the HTZs mainly apply to the Ministry of Science and Technology, whose main job is to promote scientific and technological innovation. At the same time, the ETZs are mainly audited by the Ministry of Commerce, whose main job is domestic and foreign trade and international economic cooperation. Therefore, it can be seen that the two types of DZs have more apparent differences in industrial development, with HTZs focusing more on high-tech industries and ETZs focusing mainly on manufacturing industries under foreign investment. The difference in the industry focus of the two types of DZs and the different industrial clustering within the zones will inevitably lead to differences in environmental impacts when transformed into EIPs.

## 3. Empirical Strategy and Date

We use the multi-period DID approach to estimate the overall effect of EIPs policies on the environment, setting the policy dummy variable did include: the provincial DZs variable *pdzs*, where the variable *pdzs* takes the value of 1 for municipalities with established provincial DZs, and takes the value of 0 for municipalities without established provincial DZs; the national DZs variable *ndzs*, where the variable *ndzs* takes the value of 1 for municipalities with established national DZs, and takes the value of 0 for municipalities without established national DZs; the ETZs variable *etzs*, where the variable *etzs* takes the value of 1 for municipalities with established ETZs, and takes the value of 0 for municipalities without established ETZs; the HTZs variable *htzs*, where the variable *htzs* takes the value of 1 for municipalities with established HTZs, and takes the value of 0 for municipalities without established HTZs, the EIPs variable *eips*, where the variable *eips* takes the value of 1 for municipalities with established EIPs, and takes the value of 0 for municipalities without established EIPs. Generally, this method should set one dummy variable *treated*: the cities with EIP are in the experimental group and the variable *treated* equals 1, the cities without EIP are in the control group and *treated* equals 0. In this way can the variable *treated* tell the differences between the experimental group and the control group even if there is no experiment conducted. Meanwhile, this method should set another dummy variable *post* which takes the establishment time of EIP as the dividing line. If the city has established EIP, the variable *post* is equal to 1. If not, the *post* is 0. So variable *post* can describe the differences between two periods, that is before and after the quasi-experiment. Then the regression coefficient of the interaction term *treated × post* can exactly measure the effect of EIP policy on the experimental group. However, the EIP in some cities are not established in the same year. Therefore, the typical DID method is not consistent with this study. However, it can be seen that the interaction term *treated × post* in the typical DID method is exactly the variable did in our setting [33]. The corresponding econometric model is:(1)$$pf_{it}=\beta_{0}+\beta_{1}did_{it}+\gamma X_{it}+u_{i}+\nu_{t}+\varepsilon_{it}$$

The explained variable *pf* mainly uses the emission intensity of carbon dioxide and sulfur dioxide in the city *i* at year *t*. Given the availability of data, we here use the logarithm of the amount of carbon dioxide and sulfur dioxide produced per unit of GDP for each city. The control variables *X* mainly include: the industrial structure variable *ind*, measured by the share of total value added of secondary and tertiary industries in GDP; the population density variable *pod*, measured by the number of people per unit of land area; the financial development level variable *fin*, measured by the ratio of various loan balances of financial institutions to GDP at the end of the year; the foreign direct investment variable *fdi*, measured by the ratio of total foreign direct investment to GDP, the city size variable *csz*, measured by the logarithm of the total urban population. Also, *u_i_* denotes city fixed effects, while *v_t_* denotes year fixed effects.

The sulfur dioxide emission data used are from the China City Statistical Yearbook, which are only disclosed from 2003 by this yearbook. Therefore, the data used in this paper span from 2003 to 2019, including 288 cities at the prefecture-level and above. The carbon dioxide emission data are obtained from the Center for Global Environmental Research, which provides global carbon dioxide emission data from January 2003 to December 2019. We extract the raster data within China and aggregate them according to cities to obtain the carbon dioxide emissions panel data for Chinese cities from 2003 to 2019. The statistical analysis of the main variables is shown in Table 1.

However, considering the application process of EIPs, it can be found that EIPs are transformed from DZs. The whole process goes from no DZs to the establishment of provincial DZs, from provincial DZs to national DZs, and finally from national DZs to EIPs. There are significant differences between provincial DZs, national DZs, and EIPs. The provincial DZs focus on economic growth [5,25] when provincial DZs are upgraded to national DZs, the preferential policies are increased due to the expansion in administrative level, and there are more significant benefits in terms of land rent and taxation. In contrast, national DZs are only 20 of provincial DZs and have a more substantial brand effect. On this basis, the transformation of national DZs into EIPs can enjoy the original preferences while also receiving additional preferences and subsidies. For example, local finance departments will also earmark special funds to promote the development of EIPs. Targeted financial subsidies will be provided for parks that carry out the creation of EIPs. At the same time, it can cultivate a pool of specialized talents for EIPs to provide a large number of talents. Furthermore, to promote the development of the park, the national investment policy implemented in the EIPs will be more flexible, so that more social capital and foreign investment will be attracted into the EIPs. Last but not least, the EIPs will have more opportunities for policy piloting and will enjoy more national and local preferential policies first.

When the impact of the EIPs directly analyzed without taking the upgrading and transformation process into consideration, there will be a significant error. Ignoring the transformation of DZs into EIPs and only assessing the contribution of DZs to the environment will also lead to a biased assessment of the effect of DZs. Because the contribution of EIPs is included. Meanwhile, ignoring that EIPs are upgraded from national DZs and comparing the differences between EIPs and non-EIPs in general, will also include the contribution of DZs to the environment, and result in a biased assessment of the effect of EIPs.

Therefore, this paper analyzes the role of EIPs by focusing on the whole process of policy upgrading and transformation. Firstly, the overall impact of EIPs is analyzed empirically, and cities that have established EIPs are set as the experimental group. Cities that have not established EIPs are set as the control group accordingly. And then, the upgrading process is followed to empirically analyze the effect of setting Provincial DZs with the experimental group being cities that have established Provincial DZs and the control group being cities that have not established Provincial DZs, then switching from provincial DZs to national DZs, with the experimental group being cities that have established national DZs and the control group being cities with provincial DZs that have not established national DZs. Finally, we analyze the effect of switching from national DZs to EIPs. The experimental group is cities with established EIPs, and the control group is cities with national DZs that have not established EIPs.

Further, there is a distinction between HTZs and ETZs. The industries in the HTZs include three main areas: information technology, biotechnology, and new materials technology, to drive economic growth through the development of knowledge-intensive and technology-intensive high-tech industries. However, compared to the HTZs, the ETZs aim to drive economic growth, mainly by attracting foreign investment. The industries attracted are mostly manufacturing industries. In contrast, the EIPs pay extra attention to the environment in the process of economic growth and implement the environmental regulation. The focus of the three types of parks is different in terms of their industries. The HTZs focus on high-tech industries to promote technological progress through agglomeration. While the ETZs focus on processing and manufacturing industries under foreign investment to drive the increase of total economic volume. The EIPs focus on a wider range of industries, including high-tech industries, manufacturing and processing industries and green industries to improve technology and energy utilization through recycling and agglomeration to drive economic growth and reduce pollution. Therefore, in the empirical analysis of EIPs, the differences in the types of national DZs should be considered. The environmental impacts of transforming HTZs and ETZs into EIPs should be considered separately.

The data of DZs are obtained from the “Catalogue of China Development Zones Audit Bulletin (2018 Edition)”and issued by the State Council of the People’s Republic of China, which provides both the list of provincial and national DZs. However, national DZs are upgraded from provincial DZs, and the announcement only reports the time when they became national DZs. In order to obtain the time when a provincial DZ became a provincial DZ, we first refer to the “China Development Zone Review Announcement Catalogue (2006 Edition)” to find the time when the national DZs was approved as a provincial DZs. For the DZs that were upgraded to the national level between 2007 and 2018, we use the official website to find out when they were approved as provincial DZs. The data relating to EIPs in this paper were obtained from the List of EIPs published by the Ministry of Ecology and Environment of the People’s Republic of China, which disclosed the time when the zones became EIPs.

## 4. Basic Results

First, we empirically analyze the overall impact of the overall EIPs on sulfur dioxide and carbon dioxide emissions without considering the whole upgrading process of EIPs establishment, as shown in Table 2, columns (1) and (2) present the impact of EIPs on sulfur dioxide emission intensity. Columns (3) and (4) present the impact of DZs on carbon dioxide emission intensity. The regression results are based on the variable did. Whether or not the control variable is included, EIPs will significantly reduce the intensity of carbon dioxide or sulfur dioxide emissions in the city where it is located. From the regression results of adding control variables, it can be seen that EIPs will reduce sulfur dioxide emissions by 27.8% and carbon dioxide emissions by 11.2%. When evaluating the role of EIPs, Nie et al. (2021) found that areas with EIPs reduced carbon dioxide emissions by 7.2%. It is concluded that the establishment of EIPs can promote low carbon sustainable development [41]. Song & Zhou (2021) also indicated that the establishment of EIPs reduced sulfur dioxide emissions by 25.2% in the area, which is consistent with the results of this paper [21].

This indicates that China’s EIPs can significantly reduce sulfur dioxide and carbon dioxide emissions. It also takes environmental protection into account while developing the economy. EIPs have strict environmental regulations so that high-polluting enterprises in the park, that do not meet the regulations, will withdraw from the park. At the same time, the government gives EIPs special funds and talent support. Due to the existence of environmental regulations, enterprises will also pay attention to the emission of pollutants. Therefore, companies will improve their ability to deal with pollutants to reduce pollutant emissions. Besides, the park’s circular economy model of “reduction, reuse, and recycling” can also significantly improve resource utilization efficiency and reduce resource usage and pollutant emissions. Wu (2022) pointed out that EIP policy improved the Green Innovation capacity of cities through driving mechanisms such as technological innovation, structural adjustment, and factor agglomeration effects [20]. Fan (2020) pointed out: In the EIPs, minerals made the largest reduction: 200 ha/capita, followed by fossil fuels with a reduction of 101 ha/capita which demonstrates that EIPs can reduce energy use [42]. Moreover, When EIPs attract investment, they will also give priority to green and high-tech enterprises.

In order to establish the EIP, DZs should go through the process of upgrading and transformation. We first analyze the role of no DZs to provincial DZs (excluding national DZs and EIPs) and then analyze the role of provincial DZs to national DZs (excluding EIPs). Finally, the role of the national DZs to the EIPs will be analyzed, so as to accurately analyze the impact of the EIPs.

The regression results of the variable did in Table 3 show that the emission of sulfur dioxide and carbon dioxide increases significantly from no DZs to the establishment of DZs. It is consistent with findings of Jiang (2022) and Zhao (2021) [25,33]. The regression results of the variable did in Table 4 show that if provincial DZs are transformed into national DZs (excluding EIPs), they cannot significantly reduce emissions of sulfur dioxide and carbon dioxide. However, the regression results of the variable did in Table 5 show that if national DZs are transformed into EIPs, they significantly reduce emissions of sulfur dioxide and carbon dioxide. Comparing the results in Table 3, Table 4 and Table 5, it can be seen that, only the EIPs can significantly reduce the emissions of sulfur dioxide and carbon dioxide in the entire upgrading and transformation process, The reason is that provincial DZs are established and focused on whether enterprises and industries could bring about economic growth. In order to achieve this goal, the environment could even be sacrificed. At the same time, national DZs are concerned with economic growth while also being environmentally friendly and protecting the environment to a certain extent. However, it does not carry out environmental regulation, nor does it add environmental factors to the assessment indicators. The marginal effects of environmental regulations in EIPs differ between the two zones after being transformed into EIPs that focus on environmental issues. For example, more enterprises in DZs do not meet the environmental requirements of EIPs and thus withdraw from them, resulting in more significant reductions in carbon dioxide and sulfur dioxide emissions from their zones. Meanwhile, the purpose of sustainable development of the EIPs also makes it attract more high-tech and green industries when attracting investment. Besides, the government provides special financial and talent support, so that the EIPs have a higher technical level. Under the combined effect of these factors, the DZs have significantly reduced sulfur dioxide and carbon dioxide emissions.

## 5. Heterogenous Effects in Transformation of Different Types of National DZs

As mentioned in the previous section, the HTZs and the ETZs have significant differences in their establishment objectives and leading industries, so there may be differences in the impacts on carbon dioxide and sulfur dioxide emissions after transforming into EIPs. Based on this paper, the results are shown in Table 6 and Table 7. It can be seen that the impact of transforming the ETZs into EIPs on sulfur dioxide and carbon dioxide emissions is significantly higher than the impact of transforming the HTZs. This difference is that the industries clustered in HTZs are mainly high-tech industries, Enterprises in the park have a high level of technology, resulting in fewer emissions. In contrast, the industries in the ETZs are mainly concentrated in the entire manufacturing industry, which has a higher pollution level than the HTZs. Therefore, after the transformation into a national EIPs, the environmental regulation of the park has a greater impact on the ETZs. On the one hand, more enterprises with substandard emissions are forced to leave the park compared to the HTZs. On the other hand, the resource consumption of the zone is higher, and the improvement of resource utilization rate brought about by the circular economy has a greater impact on the zone.

The previous study indicated that upgrading provincial DZs to national DZs would reduce pollutant emissions, upgrading to ETZs would not significantly affect pollutant emissions, and upgrading to national HTZs would significantly reduce pollutant emissions [13,15]. Wang and Feng (2021) indicated that the establishment of HTZs could improve the technology significantly and can also reduce sulfur dioxide emissions by 26.8% [43]. However, the transformation process of the DZs into an EIPs may be overlooked. The results in Table 6 and Table 7 show that the transformation of ETZs and HTZs into EIPs both significantly reduces sulfur dioxide and carbon dioxide emissions. The previous study may have overestimated the reduction of pollutant emissions in national DZs. This paper focuses on this transformation process and regresses the effects on carbon dioxide and sulfur dioxide emissions of the ETZs and HTZs that have not been transformed into EIPs. The results are shown in Table 8 and Table 9. Column (1) and (2) are the effects of all ETZs and HTZs on carbon dioxide and sulfur dioxide emissions, Column (3) and (4) are the effects on carbon dioxide and sulfur dioxide emissions of the ETZs and HTZs which are not transformed into EIPs. It is evident that, if all the cities with an ETZ are taken into the sample, they will not significantly affect the emissions. However, if only the cities with an ETZ are taken and without an EIP, they will aggravate the emissions, which is the real effect of ETZs. In a similar analysis for HTZs, the HTZs without EIPs have much less effect on emissions reduction than including all the HTZs with EIPs. It can be seen that the previous studies mainly get the combined effect of the DZs policy and the EIPs policy.

## 6. Parallel Trend Test

A premise of our DID strategy is that, in the absence of EIPs, the trend of the sulfur dioxide and carbon dioxide emission intensity between the treated group and the control group should be the same. Now, we construct the following model referring to Howell (2017) [44] to test the parallel trend assumption:(2)$$pf_{it}=\beta_{0}+\overset{6}{\underset{j=-6}{\sum }}\beta_{j}jl_{it}^{j}+X_{it}+u_{i}+\lambda_{t}+\varepsilon_{it}$$
where, $jl_{it}^{j}$ is the policy enactment time dummy variable. Assuming that the year when the EIPs starts to be established is *s_i_*, if *t* − *s_i_ ≤* 6, then *jl*^−6^ = 1, otherwise 0. if *t* − *s_i_* > 6, then *jl^6^* = 1, otherwise 0; if *t*-*s_i_* = j, then *jl*^j^ = 1, otherwise 0, where j = (−5, −4, −3, −2, −1, 0, 1, 2, 3, 4, 5), when *j* < 0 to reflect the control and control groups in the policy implementation before j years of policy implementation and when *j* > 0 to describes the dynamic impact of the policy after its implementation. The dummy variable *jl*^−1^ was removed from the regressions. The results are shown in Table 10, which shows that the coefficients are not significant before the policy implementation, satisfying the parallel trend. Moreover, whether in the sample of all cities or in the sample with DZs only, EIPs start to significantly reduce sulfur dioxide and carbon dioxide emissions in the third year after the policy implementation, and the emission reduction effect has been strengthened year by year.

## 7. Spatial Spillover

Based on the previous analysis, EIPs can significantly reduce carbon dioxide and sulfur dioxide emissions. However, there is still a problem: although the establishment of EIPs reduces local pollution, if EIPs are to move its own high-polluting enterprises out to the surrounding areas by absorbing low-polluting industries from other areas, then it means that the establishment of EIPs is a “self-serving” type of policy. On the other hand, if the EIPs can reduce pollution in the surrounding areas while reducing their own pollution, it can be said that the EIPs have improved their technical level and resource utilization efficiency through agglomeration and circular economy, and at the same time have the certain demonstration effect and technology spillover effect on the surrounding areas, then the policy of the EIPs is “self-benefiting and altruistic.” To answer this question, this paper empirically analyzes the spatial spillover of EIPs on sulfur dioxide and carbon dioxide emissions using a spatial double-difference approach and the corresponding econometric model is
(3)$$pf_{it}=\beta_{0}+\lambda W\times was_{it}+\beta_{1}did_{it}+\beta_{2}W\times did_{it}+u_{i}+\nu_{t}+\varepsilon_{it}$$
where *W* is a spatial weight matrix in the form of a 0-1 weight matrix, measuring the proximity of cities. $\beta_{1}$ reflects the impact of the city’s policy, while $\beta_{2}$ reflects the impact of neighboring cities’ policies (spatial spillover).

To further measure the relative magnitudes of the direct and spatial spillover effects, this paper refers to Lesage and Pace (2010) [45] and uses the partial differential approach of the spatial regression model to calculate that the direct effect reflects the impact of the EIP in the city *i* on local sulfur dioxide and carbon dioxide emissions, which includes the feedback effect: the impact of the policy in this city on other cities, in turn, affects this city. The spatial spillover effect reflects the impact of the EIP in neighboring city *j* on local sulfur dioxide and carbon dioxide emissions. The total effect is the sum of the direct effect and the spatial spillover effect, which can be interpreted as the average effect of the EIP policy of one city on sulfur dioxide and carbon dioxide emissions in all cities.

This paper empirically analyzes the spatial spillover effects of EIPs on the sample with all cities and the sample with national DZs, and the regression results are shown in Table 11. Columns (1) and (2) in Table 11 show that, EIPs not only reduce local sulfur dioxide and carbon dioxide emission intensity (as shown by the direct effect), but also have a significant spatial spillover effect, which significantly reduces the surrounding sulfur dioxide and carbon dioxide emissions (as shown by the indirect effect). However, the spillover effect of the EIPs on the full sample is significantly higher than that of the national DZs.

## 8. Intermediary Effect

The previous analysis shows that EIPs don’t reduce local environmental pollution by sacrificing the surrounding areas. Liu et al. (2015) stated that EIPs can reduce the environmental impact of development by improving the level of environmental technologies. [46] Besides, Wu & Gao (2022) also pointed out that the establishment of EIPs can significantly improve the level of innovation of enterprises in the park [20]. In order to to investigate whether EIPs reduce local pollution by improving local technology, this paper selects total factor productivity (TFP) to measure technology level to test whether EIPs reduce local environmental pollution by improving TFP as the mediating effects. And the results are shown in Table 12 and Table 13. It can be seen that the establishment of EIPs can significantly increase local TFP, but the increase range for the full sample is greater than that of national DZs, as the national DZs themselves have a higher level of TFP and a certain technology base, while the technology level of areas without national DZs is lower, which is consistent with the previous results. The results in Table 13, on the other hand, illustrate that EIPs do reduce local pollution through the increase in TFP.

## 9. Conclusions

China’s government is developing a circular economy based on industrial agglomeration through EIPs and is achieving green and high-quality development by implementing environmental regulations and improving resource utilization and reducing usage through closed-loop material circulation, multi-level energy use, and minimal waste generation. This paper illustrates the effects of the implementation of EIPs on sulfur dioxide and carbon dioxide emissions through a multi-period DID approach. Using parallel trend tests, this paper finds that EIPs begin to significantly reduce carbon dioxide and sulfur dioxide emissions in the fourth year after their establishment. Because of the characteristics of gradual transformation and upgrading of EIPs, this paper focuses on the whole process of EIPs establishment. EIPs not only have an impact on the local emissions, but also on the surrounding area, the spatial spillover of the EIPs was illustrated by using the spatial double-difference method. Finally, this paper expounds the path of EIPs affecting the environment through the mediation effect test.

Overall, EIPs can reduce sulfur dioxide and carbon dioxide emissions through environmental regulation, which encourage enterprises in the park to improve their technical level to improve the efficiency of resource use and the treatment of pollutants, indicating that it can take good care of the environment while developing the economy. At the same time, unlike previous studies, this paper finds that if we ignore the whole process of upgrading and transformation and only analyze the impact of EIP establishment, we will seriously underestimate the impact of EIPs on the emissions. EIPs are transformed from national DZs. The national DZs include HTZs and ETZs, and the transformation of HTZs and ETZs into EIPs has different environmental impacts. Compared with the HTZs, the transformation of the ETZs into EIPs has a more significant impact on sulfur dioxide and carbon dioxide emissions. The difference is that they choose to concentrate on different types of industries, with HTZs mainly concentrating on industries with low resource consumption and low environmental pollution. In contrast, ETZs mainly concentrate on manufacturing industries to attract investment, with higher sulfur dioxide and carbon dioxide emissions than HTZs. Environmental regulation makes more high-polluting enterprises in ETZs exit the park. The circular economy model brought about by the EIPs has brought about a more significant increase in resource utilization rate and a greater reduction in resource use for the industries in the ETZs, making the impact of sulfur dioxide and carbon dioxide reduction in the ETZs more significant. Therefore, it is more important to actively transform the zone into an EIPs than a HTZs, and actively introduce green industries to promote the green development of the zone in the process of transformation.

Besides, the EIPs has apparent spillover effects on the surrounding areas, reducing sulfur dioxide and carbon dioxide emissions while also reducing the emissions in the surrounding areas, which is a “self-benefiting and altruistic” type of policy. The spillover effect on all surrounding areas is more evident than on the surrounding national DZs. EIPs reduce emissions by building a circular economy rather than through the entry of high and low polluting enterprises. Thus, the EIPs provide some experience to the neighboring cities and transfer some of their knowledge and technology to the neighboring cities, indirectly reducing the emissions of the neighboring cities. However, the spillover effect of the EIPs on the neighboring countries’ DZs is not as significant as the spillover effect on all the neighboring areas because of their inherently higher technology level.

Our empirical results highlight that when a city establishes DZ, whether it is a provincial DZ or upgraded to a national DZ, it will have an adverse impact on the environment during the development process. However, when the national DZ is transformed into an EIP, it can significantly reduce pollutant emissions and improve environmental quality, at the same time improve the environmental quality of surrounding areas. Therefore, in order to achieve high-quality development, we should actively promote the transformation of DZs into EIPs instead of blindly establishing DZs. Both provincial DZs and national DZs require some degree of environmental regulation. In assessing the development of DZs, environmental indicators should be included in the evaluation, so that attention can be paid to the protection of environment in the economic development process.

## Figures and Tables

**Figure 1 ijerph-19-12957-f001:**
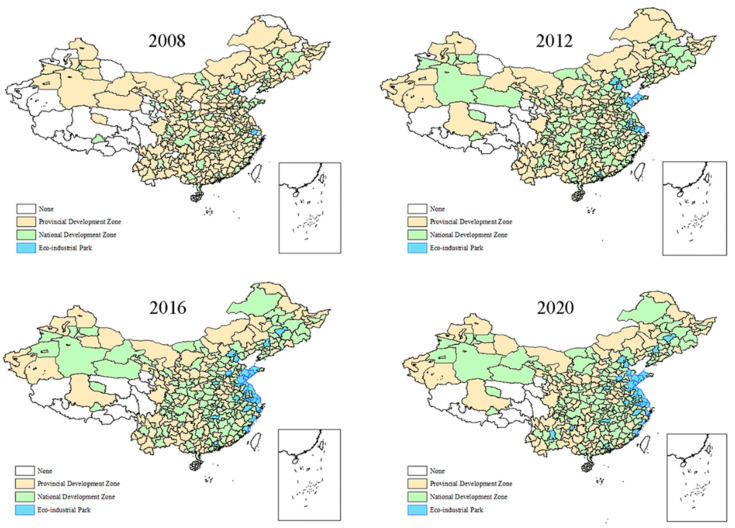
The geographic evolution of China’s EIPs.

**Figure 2 ijerph-19-12957-f002:**
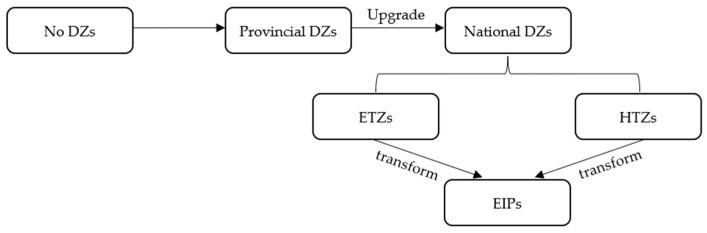
The process of upgrading and transforming DZs into EIPs.

**Table 1 ijerph-19-12957-t001:** Descriptive statistics of each variable.

Variable	Obs	Mean	Std. Dev.	Min	Max
*lns*	4627	−5.717	1.46	−14.514	−1.437
*lnco*	4830	2.921	0.813	0.425	5.441
*ind*	6714	0.837	0.104	0.387	0.999
*pod*	7333	0.43	0.335	0	3.606
*fin*	4823	0.871	0.556	0.075	9.623
*fdi*	6551	0.026	0.042	0	0.627
*csz*	7338	5.76	0.804	−3.219	8.136
*pdzs*	7830	0.87	0.336	0	1
*ndzs*	7830	0.362	0.481	0	1
*etzs*	7830	0.241	0.428	0	1
*htzs*	7830	0.262	0.44	0	1
*eips*	7830	0.026	0.16	0	1

**Table 2 ijerph-19-12957-t002:** Overall impact of EIPs on emissions.

	(1)	(2)	(3)	(4)
	*lns*	*lns*	*lnco*	*lnco*
did	−0.295 ***	−0.278 ***	−0.123 ***	−0.112 ***
	(0.054)	(0.053)	(0.015)	(0.013)
*cons*	−4.577 ***	−0.442	−12.604 ***	−9.317 ***
	(0.031)	(0.966)	(0.009)	(0.228)
Controls	NO	YES	NO	YES
City effects	YES	YES	YES	YES
Time effects	YES	YES	YES	YES
Observations	4627	4085	4772	4170
R-squared	0.792	0.767	0.846	0.891

Notes: Cluster standard error in parentheses. *** denotes *p* < 0.01. The did variable is *eips*, which demonstrates the impact of EIPs on emissions. City effects control the individual characteristics of cities. Time effects control the time trend of pollution emission which is not correlated with cities.

**Table 3 ijerph-19-12957-t003:** The impact of provincial DZs.

	(1)	(2)	(3)	(4)
	*lns*	*lns*	*lnco*	*lnco*
did	−0.182 ***	0.169 ***	0.054 ***	0.058 ***
	(0.057)	(0.058)	(0.016)	(0.014)
*cons*	−4.627 ***	−0.011	−12.513 ***	−9.594 ***
	(0.057)	(2.774)	(0.016)	(0.614)
Controls	NO	YES	NO	YES
City effects	YES	YES	YES	YES
Time effects	YES	YES	YES	YES
Observations	2574	2291	2634	2319
R-squared	0.673	0.625	0.797	0.855

Notes: Cluster standard error in parentheses. *** denotes *p* < 0.01. The did variable is *pdzs*, which demonstrates the impact on emissions after going from no development zones to the establishment of provincial DZs. City effects control the individual characteristics of cities. Time effects control the time trend of pollution emission which is not correlated with cities.

**Table 4 ijerph-19-12957-t004:** Provincial DZs to national DZs (excluding EIPs).

	(1)	(2)	(3)	(4)
	*lns*	*lns*	*lnco*	*lnco*
did	−0.012	0.021	−0.04 ***	−0.011
	(0.03)	(0.029)	(0.008)	(0.007)
*cons*	−4.514 ***	−1.189	−12.58 ***	−9.597 ***
	(0.034)	(0.976)	(0.01)	(0.234)
Controls	NO	YES	NO	YES
City effects	YES	YES	YES	YES
Time effects	YES	YES	YES	YES
Observations	4275	3767	4406	3838
R-squared	0.791	0.766	0.844	0.89

Notes: Cluster standard error in parentheses. *** denotes *p* < 0.01. The did variable changes from *pdzs* to *ndzs*, and samples with both National DZs and EIPs are excluded, which show the effect of upgrading from provincial DZs to national DZs on emissions. City effects control the individual characteristics of cities. Time effects control the time trend of pollution emission which is not correlated with cities.

**Table 5 ijerph-19-12957-t005:** National DZs to EIPs.

	(1)	(2)	(3)	(4)
	*lns*	*lns*	*lnco*	*lnco*
did	−0.18 ***	−0.279 ***	−0.119 ***	−0.133 ***
	(0.055)	(0.053)	(0.016)	(0.014)
*_cons*	−4.531 ***	2.248	−12.785 ***	−6.824 ***
	(0.059)	(1.425)	(0.018)	(0.367)
Controls	NO	YES	NO	YES
City effects	YES	YES	YES	YES
Time effects	YES	YES	YES	YES
Observations	2053	1794	2138	1851
R-squared	0.834	0.835	0.813	0.881

Notes: Cluster standard error in parentheses. *** denotes *p* < 0.01. The did variable changes from *ndzs* to *eips*, demonstrating the impact of the transition from national DZs to EIPs on emissions. City effects control the individual characteristics of cities. Time effects control the time trend of pollution emission which is not correlated with cities.

**Table 6 ijerph-19-12957-t006:** ETZs to EIPs.

	(1)	(2)	(3)	(4)
	*lns*	*lns*	*lnco*	*lnco*
did	−0.257 ***	−0.294 ***	−0.097 ***	−0.112 ***
	(0.058)	(0.057)	(0.017)	(0.015)
*_cons*	−4.577 ***	−3.433 **	−12.722 ***	−7.788 ***
	(0.068)	(1.644)	(0.021)	(0.41)
Controls	NO	YES	NO	YES
City effects	YES	YES	YES	YES
Time effects	YES	YES	YES	YES
Observations	1468	1295	1523	1334
R-squared	0.852	0.85	0.839	0.902

Notes: Cluster standard error in parentheses. ** denotes *p* < 0.05, and *** *p* < 0.01. The did variable is *eips* and the full sample is the city where ETZs are established, demonstrating the impact on emissions after the transition from ETZs to EIPs. City effects control the individual characteristics of cities. Time effects control the time trend of pollution emission which is not correlated with cities.

**Table 7 ijerph-19-12957-t007:** HTZs to EIPs.

	(1)	(2)	(3)	(4)
	*lns*	*lns*	*lnco*	*lnco*
did	−0.125 **	−0.217 ***	−0.118 ***	−0.132 ***
	(0.063)	(0.058)	(0.019)	(0.016)
*cons*	−4.571 ***	7.58 ***	−12.883 ***	−5.395 ***
	(0.067)	(2.182)	(0.021)	(0.601)
Controls	NO	YES	NO	YES
City effects	YES	YES	YES	YES
Time effects	YES	YES	YES	YES
Observations	1433	1261	1499	1303
R-squared	0.845	0.86	0.817	0.879

Notes: Cluster standard error in parentheses. ** denotes *p* < 0.05, and *** *p* < 0.01. The did variable is *eips* and the full sample is the city where HTZs are established, demonstrating the impact on emissions after the transition from ETZs to EIPs. City effects control the individual characteristics of cities. Time effects control the time trend of pollution emission which is not correlated with cities.

**Table 8 ijerph-19-12957-t008:** Provincial DZs to ETZs.

	The Cities with EIPs		The Cities without EIPs	
	(1)	(2)	(3)	(4)
	*lns*	*lnco*	*lns*	*lnco*
did	0.032	−0.009	0.087 ***	0.003
	(0.037)	(0.009)	(0.031)	(0.008)
*cons*	−5.479 ***	−10.948 ***	−1.168	−9.606 ***
	(1.49)	(0.348)	(0.974)	(0.234)
Controls	YES	YES	YES	YES
City effects	YES	YES	YES	YES
Time effects	YES	YES	YES	YES
Observations	2682	2723	3767	3838
R-squared	0.698	0.885	0.767	0.89

Notes: Cluster standard error in parentheses. *** *p* < 0.01. The did variable is *htzs* and the full sample is the city where Provincial DZs are established, demonstrating the impact on emissions after the transition from Provincial DZs to ETZs. City effects control the individual characteristics of cities. Time effects control the time trend of pollution emission which is not correlated with cities.

**Table 9 ijerph-19-12957-t009:** Provincial DZs to HTZs.

	The Cities with EIPs		The Cities without EIPs	
	(1)	(2)	(3)	(4)
	*lns*	*lnco*	*lns*	*lnco*
did	−0.174 ***	−0.045 ***	−0.095 ***	−0.03 ***
	(0.045)	(0.011)	(0.035)	(0.009)
*_cons*	3.564	−10.88 ***	−3.258 ***	−9.675 ***
	(2.747)	(0.621)	(0.263)	(0.234)
Controls	YES	YES	YES	YES
City effects	YES	YES	YES	YES
Time effects	YES	YES	YES	YES
Observations	2648	2692	4242	3838
R-squared	0.714	0.87	0.76	0.89

Notes: Cluster standard error in parentheses. *** denotes *p* < 0.01. The did variable is *htzs* and the full sample is the city where Provincial DZs are established, demonstrating the impact on emissions after the transition from Provincial DZs to HTZs. City effects control the individual characteristics of cities. Time effects control the time trend of pollution emission which is not correlated with cities.

**Table 10 ijerph-19-12957-t010:** Parallel Trend Tests and Dynamic Effects.

	All the Cities		Only Cities with a National DZ	
	(1)	(2)	(3)	(4)
	*lns*	*lnco*	*lns*	*lnco*
pre_6	0.17 *	0.055 *	0.224 **	0.07 **
	(0.103)	(0.031)	(0.096)	(0.032)
pre_5	0.071	0.047	0.128	0.048
	(0.129)	(0.04)	(0.117)	(0.04)
pre_4	0.035	0.046	0.07	0.043
	(0.129)	(0.04)	(0.116)	(0.039)
pre_3	0.011	0.031	−0.008	0.024
	(0.13)	(0.04)	(0.115)	(0.039)
pre_2	0.004	0.022	−0.012	0.016
	(0.13)	(0.04)	(0.114)	(0.039)
current	−0.176	−0.029	−0.2 *	−0.036
	(0.129)	(0.04)	(0.111)	(0.038)
post_1	−0.05	−0.045	−0.09	−0.051
	(0.135)	(0.04)	(0.117)	(0.038)
post_2	−0.07	−0.051	−0.104	−0.057
	(0.134)	(0.04)	(0.116)	(0.038)
post_3	−0.176	−0.083 **	−0.179	−0.083 **
	(0.138)	(0.04)	(0.119)	(0.038)
post_4	−0.349 **	−0.112 ***	−0.337 ***	−0.117 ***
	(0.146)	(0.043)	(0.127)	(0.041)
post_5	−0.298 **	−0.123 ***	−0.326 **	−0.127 ***
	(0.147)	(0.044)	(0.127)	(0.042)
post_6	−0.397 ***	−0.165 ***	−0.401 ***	−0.168 ***
	(0.13)	(0.037)	(0.117)	(0.037)
cons	−0.477	−12.61 ***	2.546 *	−12.806 ***
	(0.966)	(0.01)	(1.428)	(0.02)
Observations	4085	4772	1794	2138
R-squared	0.768	0.846	0.838	0.815

Notes: Cluster standard error in parentheses. * denotes *p* < 0.1, ** *p* < 0.05, and *** *p* < 0.01.

**Table 11 ijerph-19-12957-t011:** Spatial spillover.

	All the Cities		Only Cities with a National DZ	
	(1)	(2)	(3)	(4)
	*lns*	*lnco*	*lns*	*lnco*
*Main*	−0.1833 ***	−0.0723 ***	−0.1288 **	−0.1228 ***
did	(0.057)	(0.012)	(0.063)	(0.017)
*Wx*	−0.1397	0.0026	0.059	0.0158
did	(0.085)	(0.019)	(0.08)	(0.023)
Spatial	0.2220 ***	0.5980 ***	0.0933 **	0.4019 ***
rho	(0.018)	(0.013)	(0.036)	(0.037)
Direct Effects	−0.1909 ***	−0.0797 ***	−0.1245 *	−0.1288 ***
	(0.057)	(0.013)	(0.064)	(0.018)
Spillover Effects	−0.2233 **	−0.0931 **	0.0307	−0.0287 *
	(0.101)	(0.04)	(0.051)	(0.017)
Total Effects	−0.4142 ***	−0.1728 ***	−0.0938	−0.1575 ***
	(0.108)	(0.046)	(0.075)	(0.028)
N	4743	4743	1020	1020
R-squared	0.1159	0.2391	0.0124	0.1444

Notes: Cluster standard error in parentheses. * denotes *p* < 0.1, ** *p* < 0.05, and *** *p* < 0.01. The did variable is *eips*, showing the effect of EIPs on the whole sample and the sample of national DZs.

**Table 12 ijerph-19-12957-t012:** Impact of Ecoregion Establishment on TFP.

	All the Cities		Only Cities with a National DZ	
	(1)	(2)	(3)	(4)
	*tfp*	*tfp*	*tfp*	*tfp*
did	0.049 ***	0.051 ***	0.039 ***	0.046 ***
	(0.012)	(0.011)	(0.01)	(0.01)
*cons*	0	−1.691 ***	0.347 ***	−3.403 ***
	(0.006)	(0.18)	(0.01)	(0.253)
Controls	NO	YES	NO	YES
City effects	YES	YES	YES	YES
Time effects	YES	YES	YES	YES
Observations	4260	3882	1745	1608
R-squared	0.004	0.142	0.02	0.21

Notes: Cluster standard error in parentheses. *** denotes *p* < 0.01. The did variable is *eips*, demonstrating the effect of *E*IP*s* on TFP. City effects control the individual characteristics of cities. Time effects control the time trend of pollution emission which is not correlated with cities.

**Table 13 ijerph-19-12957-t013:** Intermediary effect.

	All the Cities		Only Cities with a National DZ	
	(1)	(2)	(3)	(4)
	*lns*	*lnco*	*lns*	*lnco*
*tfp*	−0.685 ***	−0.828 ***	−1.469 ***	−0.793 ***
	(0.091)	(0.016)	(0.153)	(0.028)
did	−0.147 **	−0.026 **	−0.119 **	−0.035 ***
	(0.06)	(0.01)	(0.056)	(0.01)
*cons*	−1.638	−12.055 ***	−5.042 ***	−11.571 ***
	(0.996)	(0.17)	(1.534)	(0.283)
Controls	YES	YES	YES	YES
City effects	YES	YES	YES	YES
Time effects	YES	YES	YES	YES
Observations	3827	3856	1580	1594
R-squared	0.718	0.888	0.801	0.898

Notes: Cluster standard error in parentheses. ** denotes *p* < 0.05, and *** *p* < 0.01. The did variable is *eips*, demonstrating that EIPs reduces carbon dioxide and sulfur dioxide emissions by increasing TFP. City effects control the individual characteristics of cities. Time effects control the time trend of pollution emission which is not correlated with cities.

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
