# Peer review of "Does China’s National Demonstration Eco-Industrial Park Reduce Carbon Dioxide and Sulfur Dioxide—A Study Based on the Upgrading and Transformation Process"

_ijerph, 2022, doi:10.3390/ijerph191912957_

Round 1
Reviewer 1 Report
This paper reflects a great piece of work. The environmental impacts of China's National Demonstration Eco-industrial Parks are very interesting. I think the article should be a little bit improved, especially its clarity.
First, to increase the clarity of the approval process, I suggest making a figure/scheme, for example, in section "2. Background" somewhere near line 213. For many people in different regions of the world, less familiar with Development Zones, National and Provincial levels Development Zones, and Eco-industrial Parks, the description of this procedure may not be clear.
Please consider whether the keyword "Upgrading and Transformation" is too general. Maybe better is "Development Zones Transformation".
Figure 1. The geographic evolution of China's EIPs is hardly readable. I suggest improving it. Additionally, it is worth adding a few sentences about the evolutionary distribution shown in figure1, for example, starting in line 181.
I do not know if conclusions are necessary in the Introduction section. Maybe better would add a description of the paper structure.
Equation 1. line 253. The value of the variable did depend on EIPs. If did depends on DZs also, it should be added in lines 252-253.
Is it possible to describe the models (equations) describing the transformation shown in tables 4-9? In my opinion, it helps better and quicker understand results.
Why did you use in calculations the logarithm of the amount of carbon dioxide and sulfur dioxide produced per unit of GDP (lines 255-256) only? Usually, we use a natural logarithm of all variables in tests/calculations.
Table 1 should describe all variables, including DZs (national, provincial) and HTZs.
Reading the text, I had an impression that some information was repeated, for example, in lines 366-372 and earlier description of the approval process. Maybe it will be better to refer to the Figure with this process in this part of the paper (if it will be added).
The discussion referring to the literature is necessary in section "4. Basic Results" and the following.
In the "Conclusions" section, there is one clear sentence with a policy recommendation: "Therefore, in order to achieve high-quality development, we should actively promote the transformation of DZs into EIPs instead of blindly establishing DZs." Please add more policy recommendations.
Reviewer 2 Report
This paper presents one of the most complete descriptions of the development and transformation process of EIPs in China that I have read. There is some duplication of information in the text regarding the process which should be addressed. I am concerned about the statistical analysis conducted and wonder whether it is all needed to draw the conclusions.
I do have an extensive number of points for the authors to consider:
The paragraph from lines 91-110 is important but confusing.
Lines 110-111 state that the paper" empirically investigates the environmental impacts of EIPs." This is not accurate. The paper investigates emissions of SO2 and CO2 only but does not actually address the impacts of these emissions.
Line 114 " such a selection process will inevitably lead to spatial flows of resources and pollutants " is an overstatement without support.
Lines 125-126. You are not actually analyzing the impacts of the emissions on the surrounding environment eg on health, biota, water bodies etc.
Lines 132-133. Needs to be clarified.
Lines 136-145. I would not include conclusions in the introduction.
Lines 154-161. When considering reduction and recycling, one has to separate the efficiency improvements in industrial operations. Many industrial recycling claims are actually the result of efficiency improvements for financial benefit.
Figure 1 is not very clear to this reader.
Lines 184-218 is an excellent description of the EIP development process.
Line 230. What is "scarce business card"?
Lines 205-209. A listing of the eco-environmental protection indicators and ecological improvement potential would be helpful to see where SO2 and CO2 fit.
Lines 249. The authors must do better at describing how and why the multiperiod DID and equation (1) "estimates the overall effect of EIP policies on the environment" This is quite a claim.
I have reviewed many papers by Chinese researchers over a long period of time and there is a tendency to use obscure statistical methods to prove a hypothesis that should be tested by simpler means. I ask that you consider whether simpler methodology could have arrived at the same conclusion?
Lines 301-311. It is not simply a matter of having an EIP when selecting experimental and control groups. Should not the cities be of comparable size with a similar industrial base?
Table 2 actually uses the term impact appropriately.
In tables 3 to 5, the terms city effects and time effects require more explanation, perhaps as footnotes.
Lines 453-467 seem unnecessarily complicated. Also what is "policy shock"?
Lines 472-480 suggest some responsibility on the part of EIP administrators for reducing pollution in surrounding areas. Would this not be the responsibility of city or provincial administrators?
Section 8 seems suspect to me. There is no indication that a survey of industries was conducted to assess the technologies used within and outside the EIP.
Table 2. What is an "eco-region"? This term has a particular meaning in biological litetrature.
Conclusions: Lines 539-541 about HTZs and ETZs are not clear.
And finally some of the statements about resource utilization are not supported by the research.
Round 2
Reviewer 2 Report
This revision is an improvement over the original version. The title continues to suggest the paper discusses environmental impacts. It only addresses CO2 and SO2 emissions and not the environmental impacts of those emissions. This has to change.
Line 94-95. I do not understand how previous studies which did not address upgrading and transformation might have caused bias !!!
Line 117. What are inter-regional flows of companies? Are you referring to the movement of companies from one region into an EIP in another region? And what is a region in the context of China?
Line 177-178 ".... may increase the economic efficiency to a certain extent due to the increase in efficiency".???? In the latter are you referring to material efficiency?
Lines 396 -402. It seems that other studies have already concluded that EIPs result in reductions of CO2 and SO2? What is it about upgrading and transformation that makes a big difference?
Line 399. What is "low carbon health development"? This is a new term.
Lines 404-407. The notes here and in other tables. do not explain things well.What does "time effects control the time characteristics of cities" refer to?
Lines 420-421, What is the indicator xE + yha/capita? What does it actually tell the reader?
Lines 473-474. "It can be seen.... that transformation into an EIP results in reduced CO@ and SO2 compared to HTZs." IS this really surprising given what other authors have reported? But then lines 487-490 seem to contradict this.
Line 490. Table 10 is a significant finding that should be highlighted in the conclusion.
Section 7 is also a significant finding and perhaps one of the more important in the paper. But what does this have to do with upgrading and transformation?
Even after reading the revised version, I am not convinced by the statement that"if we ignore the whole process of upgrading and transformation we will seriously underestimate the impact of EIPs on the environment" . The premise of the paper needs strengthening to make this case. I do continue to feel that the description of the transformation process of existing DZs into other forms and ultimately EIPs is very helpful.
